# Woodfuel Consumption in Refugee Hosting Areas and Its Impact on the Surrounding Forests—The Case of Uganda

**Arturo Gianvenuti \*, Nelly Grace Bedijo, Rashed Jalal**  **, Leonidas Hitimana, Sven Walter, Thais Linhares-Juvenal and Zuzhang Xia**

Food and Agriculture Organization of the United Nations (FAO), 00153 Rome, Italy
\* Correspondence: arturo.gianvenuti@fao.org

**Abstract:** The high dependency of households on woodfuels in Uganda is a renowned driver of forest degradation. Refugee settlements might aggravate the pressure on the environment caused by woodfuel consumption in the absence of impact assessments and measures to improve environmental management and build sustainable livelihoods. In collaboration with the Government of Uganda, UNHCR, and the World Bank, FAO conducted assessments on woodfuel consumption at the household level in displacement settings in Uganda, as well as its impact on the surrounding forests, through field forest inventories, household surveys, and geospatial analysis. The results of these assessments highlight the impact of woodfuel consumption and other drivers on forest degradation, and provide guidance for the development of comprehensive interventions on landscape management and improvement of household cooking fuels and technologies, targeting both refugee and host communities.

**Keywords:** refugee settlements; Uganda; woodfuel; forest degradation; energy; livelihoods



## 1. Introduction

Ongoing regional instability has led to the forced displacement of more than 1.5 million refugees and asylum seekers to Uganda [1], mostly from South Sudan and the Democratic Republic of the Congo (DRC), making Uganda the largest refugee host country in Africa and the third largest in the world [2]. This has resulted in the establishment or reopening of some of the world's largest refugee settlements.

The country experienced a significant increase in refugee numbers starting from the second half of 2016, which resulted from an inflow of refugees from South Sudan [3]. This was followed, starting in early 2018, by an additional inflow of refugees from DRC [4].

These displacements have often been accompanied by environmental impacts, such as land degradation and forest depletion, due to increased demand for woodfuel for cooking and timber [5,6]. Deforestation and forest degradation are not new phenomena in Uganda [7], but the refugee presence can add to existing pressures on the environment and potentially become a source of tension between refugees and host communities competing for the use of natural resources [8]. While deforestation involves the conversion of forest to another land use or the long-term reduction of tree canopy cover below the 10 percent threshold [9], forest degradation refers to a process that leads to a temporary or permanent deterioration in the density or structure of vegetation cover or its species composition that leads to a lower productive capacity [10]. In Uganda, forest resources play a key role in supporting livelihoods. Over 90 percent of the Ugandan population also rely on woodfuel, such as firewood and charcoal, as the most common source of energy for cooking [11].

Coordinated assessments in partnership with humanitarian actors are key to provide the evidence base for the environmental impact, and to identify the needs of the affected population. Assessment of the sustainability of woodfuel extraction in humanitarian situations can be particularly challenging due to the limited availability of basic data, such

as woody biomass stocks, harvesting methods, population census, energy consumption, and energy needs [12]. In addition, the relationship between supply and demand of woodfuel is often embedded in complex systems that include external factors of a non-forestry nature, which influence the capacity to provide a forestry-based solution [13].

Successful integration of woodfuel supply and demand assessment has been achieved by using the Woodfuel Integrated Supply/Demand Overview Mapping (WISDOM) methodological approach, which was developed in order to quantify spatially explicit imbalances between supply and demand for woody biomass areas, which face the most acute problems [14]. Building on this approach, FAO and UNHCR have developed a methodology for assessing woodfuel supply and demand in displacement settings, which can be used as an entry point for identifying appropriate forestry interventions, as well as supporting local forest resource management and planning [12].

Between 2018 and 2019, FAO, in collaboration with the Government of Uganda, UNHCR, and the World Bank, conducted assessments on woodfuel supply and demand in the refugee hosting areas in northern, western, and southwestern Uganda, as well as and its impact on the surrounding forests, through field forest inventories, household surveys, and geospatial analysis. A series of interventions have been recommended to mitigate existing pressure on forest resources, enhance sustainable woodfuel supply and demand, and contribute to resilience-building of both refugees and host communities.

This study re-examined these assessments with a focus on the woodfuel supply and demand in displacement settings across the country, and on the associated impacts on forest resources.

## 2. Methodology

The assessments involved a combination of a desk review, field survey, and remote sensing analysis. The field survey comprised assessment of woodfuel consumption and associated challenges in selected refugee settlements and host community villages, as well as a study of biophysical parameters of woodlands and bushlands in preselected hotspots.

These assessments build on the methodology developed in the joint FAO–UNHCR technical handbook, Assessing Woodfuel Supply and Demand in Displacement Settings [12]. The methodology comprised three components: (1) assessment of woodfuel demand and associated challenges; (2) assessment of woodfuel supply, including above-ground biomass (AGB) stock, land cover classification, and changes; and (3) identification of interventions to address issues related to energy access, forest resource degradation, and livelihoods.

### 2.1. Study Sites

The area of interest (AoI) for this study includes a 'buffer zone' up to 5 km from the boundaries (encompassing the area within a reasonable walking distance that is most likely to serve as a woodfuel source) of 14 settlements in the north of Uganda, and 6 settlements in the west and southwest (Table 1, Figures 1 and 2). A wider AoI up to 15 km away was also assessed in order to understand trends and dynamics within host communities.

Having considered biophysical and socioeconomic characteristics, such as the extent of tree cover loss, the main land use and land cover (LULC) classes, agro-ecological zones, the presence of protected areas, and the size of the settlements in terms of population, the refugee settlements of Bidibidi, Maaji, Kyaka II, and Kyangwali were purposefully selected to carry out household surveys and field forest inventory.

**Table 1.** Refugee settlements included in the assessments.

|  | Settlement Name | District | Establishment Date |
|---|---|---|---|
| **Refugee settlements in northern Uganda** | | | |
| 1 | Bidibidi | Yumbe | August 2016 |
| 2 | Imvepi | Arua | February 2017 |
| 3 | Rhino extension—Omugo | Arua | January 2017 |
| 4 | Agojo | Adjumani | January 2016 |
| 5 | Ayilo I | Adjumani | January 2015 |
| 6 | Ayilo II | Adjumani | July 2014 |
| 7 | Boroli I/II | Adjumani | January 2014 |
| 8 | Maaji I [a] | Adjumani | January 1997 |
| 9 | Maaji II [a] | Adjumani | January 1997 |
| 10 | Maaji III [a] | Adjumani | January 1997 |
| 11 | Nyumanzi | Adjumani | January 2014 |
| 12 | Pagirinya | Adjumani | January 2016 |
| 13 | Palorinya | Moyo | December 2016 |
| 14 | Palabek | Lamwo | April 2017 |
| **Refugee settlements in western and southwestern Uganda** | | | |
| 15 | Kyaka II | Kyegegwa | 2005 |
| 16 | Kyangwali | Kikuube | 1960 |
| 17 | Rwamwanja | Kamwenge | 1964; closed 1995; reopened 2012 |
| 18 | Kiryandongo | Kiryandongo | 1990; closed 1996; reopened 2014 |
| 19 | Nakivale | Isingiro | 1960 |
| 20 | Oruchinga | Isingiro | 1961 |

Note: [a]. Settlements established in 1997 and reopened in 2015.

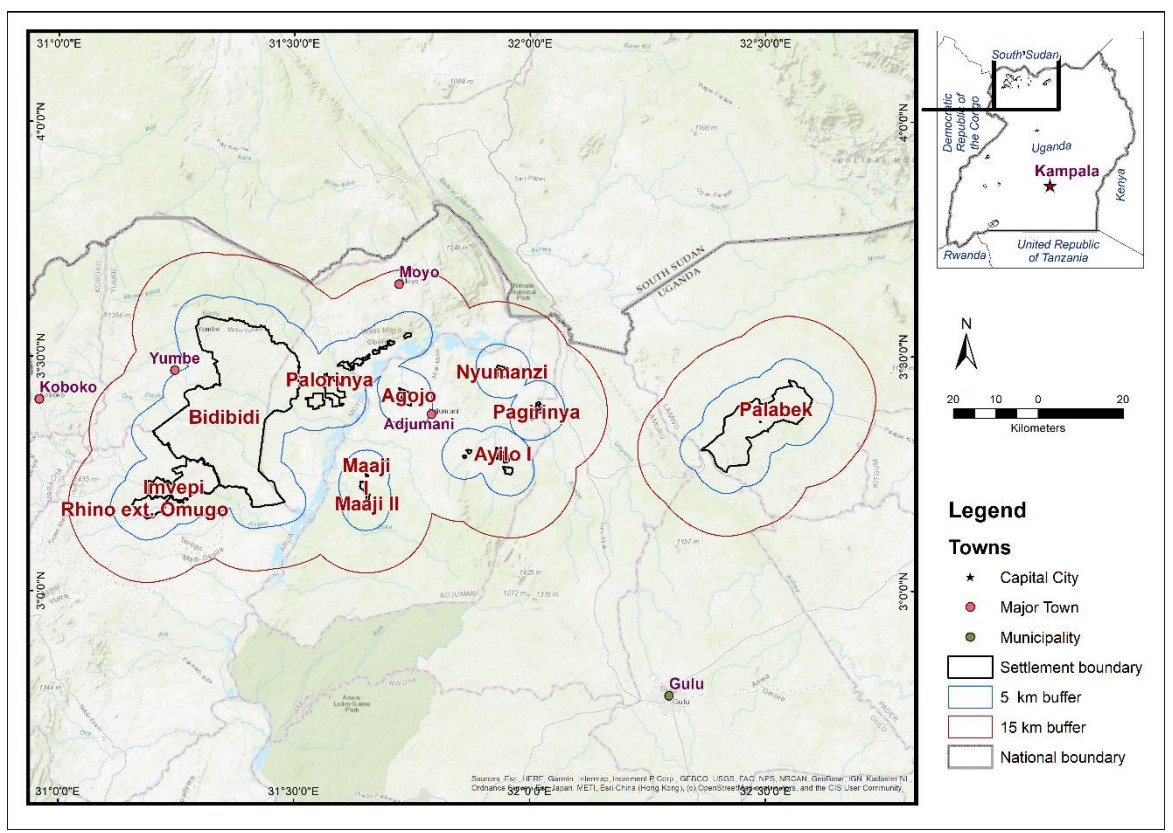

**Figure 1.** Target refugee settlements and areas of interest in northern Uganda.

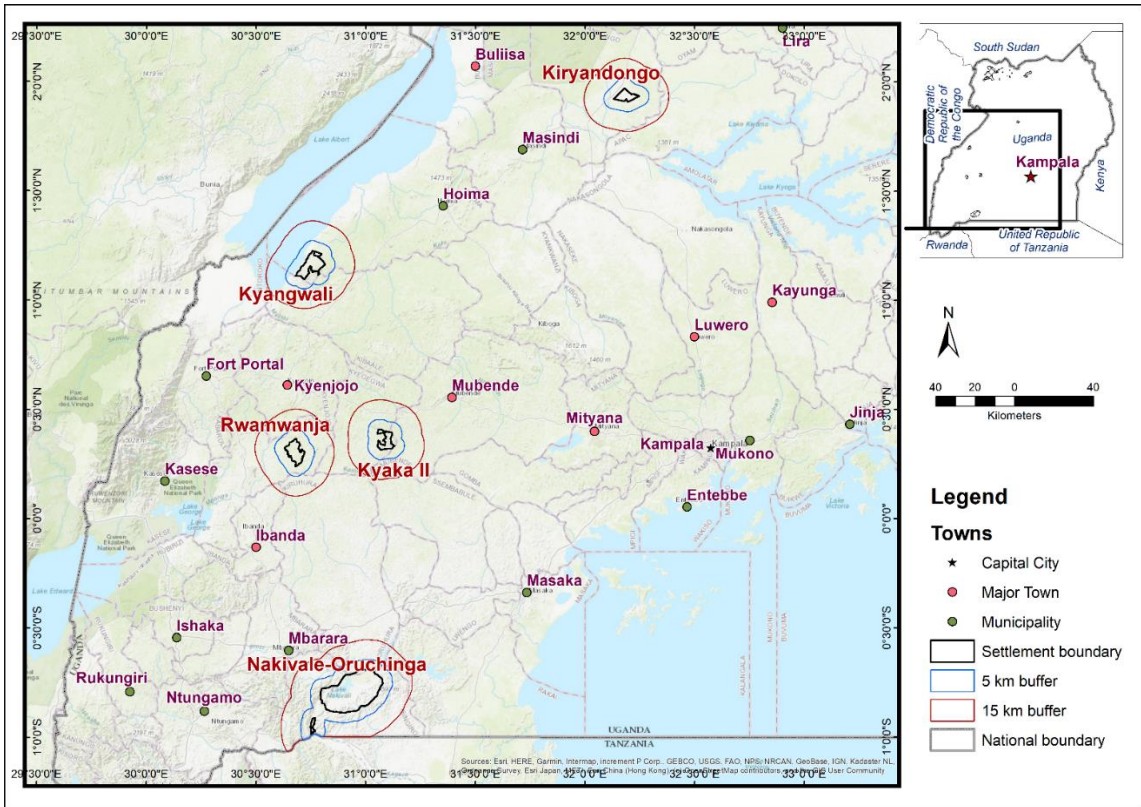

**Figure 2.** Target refugee settlements and areas of interest in western and southwestern Uganda. Note: The boundaries, names, and designations on these maps do not imply official endorsement or acceptance by the United Nations.

*2.2. Woodfuel Data Collection and Analysis*

The assessment in the north of Uganda took place in 2018, while the assessment in the west and southwest in 2019. A total of 1005 refugee and host community households were surveyed in four selected refugee settlements and surrounding hosting areas. More specifically, in northern Uganda, the survey was carried out for 174 refugee households in the Bidibidi settlement (Yumbe District) and the Maaji settlement (Adjumani District), as well as in 168 households in host communities in the Ciforo (Adjumani District) and Okangali (Yumbe District) sub-counties. Meanwhile, in western and southwestern Uganda, 177 refugee households were surveyed in the Kyangwali settlement (Kikuube District), 193 in the Kyaka II settlement (Kyegegwa District), and 293 host community households in six surrounding villages. This sample size has been designed taking into account a two-stage sample selection with an overall error of maximum 0.05 and a confidence level of 95 percent.

A quantitative household questionnaire and qualitative interviews in the refugee and host communities generated information on energy consumption for cooking, average time spent by households to collect firewood, types of cooking systems used, associated challenges, and related livelihood issues. Systematic sampling was employed for the selection of households in each location.

The total woodfuel consumption figures for each settlement are averages extrapolated from the household survey data of the four surveyed AoIs: Maaji in Adjumani District, Bidibidi in Yumbe District, Kyaka II in Kyegegwa District, and Kyangwali in Kikuube District. The average consumption of Maaji and Bidibidi are extended to the other refugee settlements in northern Uganda, while Kyaka II consumption data are extended to Rwamwanja, and the Kyangwali figures to Kiryandongo, as they are in the same agro-ecological zones [15]. No data were extrapolated to Nakivale and Oruchinga refugee

settlements, these belonging to a quite different social and ecological context. Total wood-fuel consumption takes into account both firewood (expressed on an air-dry basis) and charcoal (expressed in firewood equivalent, assuming a conversion efficiency of 20 percent).

### 2.3. Biophysical Field Inventory and Analysis

In northern Uganda, biophysical field data were collected to estimate biomass stocks for the following five LULC classes: woodland, bushland, cropland, woodland degraded, and bushland degraded (Table 2). Degraded classes refer to a partial removal of vegetation.

**Table 2.** LULC classes and AGB stock data used in the AoIs in northern Uganda.

| LULC Class | No. of Plots | Average AGB (t/ha) | Brif Description |
| --- | --- | --- | --- |
| Woodland | 15 | $38 \pm 7$ | Trees and shrubs (average height > 4 m) |
| Bushland | 10 | $27.8 \pm 5$ | Bushes, thickets, shrubs (average height < 4 m) |
| Cropland | 21 | $9.14 \pm 5.23$ | Mixed farmland, small holdings, in use or recently used, with or without trees |
| Degraded woodland | 7 | $25.3 \pm 18.5$ | Woodland with partial removal of vegetation |
| Degraded bushland | 14 | $3.94 \pm 3.95$ | Bushland with partial removal of vegetation |

The LULC classification is part of Uganda's national mapping system, and was used in these assessments to gain a better understanding of the dominant LULC classes in the AoIs.

Since the focus of the assessment was on LULC classes with potential woodfuel resources, grasslands were not considered in the first assessment in northern Uganda, as they contain very low AGB. Originally planned to be included in the biophysical survey, tropical high forests (THFs) were ultimately excluded, as their location was found to be too remote for refugees to access, situated 10 km south of the Maaji settlements (Adjumani District).

A total of 95 plots were initially identified in the targeted area in the north through a statistical stratified random sampling approach, including the THF class. Plot allocation targeted an equal distribution across classes (15 plots per class regardless of the area proportion) and ensured that rare classes (in particular, degraded woodland and degraded bushland) were well represented. A preassessment of the plots was carried out using Collect Earth, a free and open-source software program developed by FAO for land monitoring, in order to validate their land cover type and the tree cover loss for the degraded land cover classes, as well as to reach the target sample number for each stratum. In the field survey in northern Uganda, a total of 67 out of the initial 95 plots were measured, due to problems in accessing some plots and for the exclusion of the THF class in the field survey.

At each sampling location, a circular plot of 0.05 ha (12.6 m radius) was established (Figure 3). Within the first quadrant of the plot (between points 2 and 3), shrubs were measured (including basal diameter, crown diameter and average height, and number of stems [in the case of clustered shrubs]). All standing trees (alive and dead) of at least 3 cm diameter at breast height (DBH) were also measured in the first quadrant. In the rest of the plot, the minimum measured DBH was 5 cm. Other tree parameters recorded were species and total height. Within a smaller radius of 4 m (giving a circle of 0.01 ha), all saplings and deadwood were measured.

AGB was calculated using the allometric equations of Chave et al. [16], which were also used in Uganda's National Biomass Study (NBS) [15]. R scripts, developed for REDD+ (Reducing Emissions from Deforestation and Forest Degradation) in the NBS to support analyses and make summary statistics from field inventories, were used to estimate stocks.

Plot-level results were aggregated into LULC classes, as assigned to plots during the field inventory (Table 2). The woody biomass from shrubs was estimated using the NBS equation for small trees.

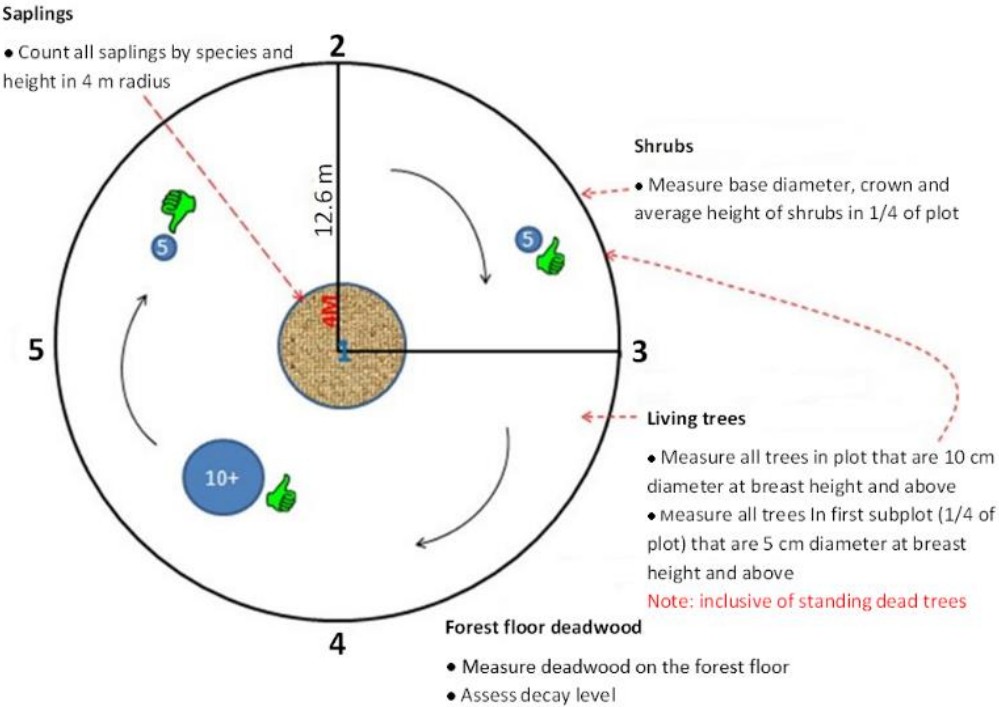

**Figure 3.** Plot design for the biophysical inventory. Source: FAO & UNHCR, 2017 [17].

In western and southwestern Uganda, AGB stocking data were provided by the National Forestry Authority (NFA) as results of a recently conducted field forest inventory which targeted the same AoI. These data were used for each land cover class (Table 3).

**Table 3.** LULC classes and AGB stock data used in the AoIs in western and southwestern Uganda.

| Class in LULC Map | Average AGB (t/ha) |
|---|---|
| Plantations, broad-leaved | 90.6 |
| Plantations, coniferous | 53.9 |
| THF well-stocked | 273.7 |
| THF low-stocked | 127.6 |
| Woodland | 12.6 |
| Bushland | 7.6 |
| Grassland | 5.3 |
| Wetland | 1.6 |
| Subsistence farmland | 10.1 |
| Commercial farmland | 10.1 |
| Built-up areas | 4.1 |
| Water | 0.0 |
| Impediment (bare soil, bare rock, and so on) | 0.7 |

Average annual biomass growth data for the various LULC classes in each of Uganda's agro-ecological zones were obtained from the NBS [15] in order to consider the annual AGB

increment in the integration of woodfuel supply and demand for each refugee settlement, and to assess the annual deficit of biomass stock. National averages were used for western and southwestern Uganda, and the values for the moist lowland zone were used for the targeted settlements in northern Uganda (Table 4).

**Table 4.** Biomass growth annual increment (air-dry matter) for selected LULC classes, as national averages and for semi-moist lowlands.

| LULC Class | National Averages (t/ha) | Semi-Moist Lowland (t/ha) |
| --- | --- | --- |
| Built-up areas | 3 | - |
| Bushland | 1 | 0.3 |
| Grassland | 1 | 1 |
| Plantations, broad-leaved | 13 | - |
| Subsistence farmland | 1 | 1.4 |
| THF well-stocked | 15 | - |
| THF low-stocked | 11 | - |
| Woodland | 5 | 3.6 |

*2.4. Biomass Stock Changes*

The distribution of woody biomass was mapped using remote sensing. In the AoIs in northern Uganda, biomass stock changes were assessed for 2010–2013 ("before South Sudan crisis") and 2014–2018 ("after South Sudan crisis"). AGB stock values were applied to each LULC class considered for northern Uganda (Table 2) in order to estimate and map the biomass stock. In order to provide spatially and temporally explicit information on changes in biomass over time in the target AoIs in northern Uganda, a time series approach was employed by running the Breaks for Additive Seasonal and Trend (BFAST) algorithm [18–21] and incorporating available Landsat satellite imagery from the U.S. Geological Survey.

The results of the BFAST algorithm were reclassified into loss and degradation maps for the two periods of interest, and were overlaid to the 2010 and 2015 LULC maps, respectively, to determine whether changes occurred in each land cover type.

LULC maps for 2010 and 2015 were reclassified. The original thirteen land cover classes identified were reduced to four based on their prominence in the landscape, accessibility, and biomass content: 1. woodland, 2. bushland, 3. cropland, and 4. other. The classes of the land cover maps were combined with the 'degradation/loss mask' obtained from BFAST for the two periods. In more detail, degraded classes refer to a partial removal of vegetation, while loss occurs when there is complete vegetation removal. For these last classes, woody biomass is assumed to be zero.

The parameters used to run BFAST for this analysis were as follows:

- For the changes between 2010 and 2013:

  ○ Beginning of historical period: 1 January 2005
  ○ Beginning of monitoring period: 1 January 2010
  ○ End of monitoring period: 31 December 2013

- For the changes between 2014 and 2018:

  ○ Beginning of historical period: 1 January 2010
  ○ Beginning of monitoring period: 1 January 2014
  ○ End of monitoring period: 16 April 2018

The output of the time series analysis is "magnitude" of change. Magnitude can vary from strongly negative (for example, deforestation) to strongly positive (for example, reforestation or revegetation). Classification of magnitude values requires creating thresholds to distinguish change classes and to create classes capable of being summarized and mapped. In order to relate "magnitude" values obtained in the analysis with on-the-ground change, the results need to be calibrated based on reliable data. Results in this study were calibrated with field-based observations and very high spatial resolution imagery from Google Earth and Worldview 2, /3, as well as GeoEye1 imagery provided by the United Nations Institute for Training and Research (UNITAR). The results also took into account the household survey on woodfuel consumption.

The processing generated a three-band raster dataset covering the AoIs in northern Uganda, where the date of break and the magnitude of detected change are recorded for each pixel (band 1 and band 2 of the resulting output). In order to identify the changes within the AoI, the layer of change magnitude was used. This is computed as the median residual ("difference or distance") between the predicted and observed values within the monitoring period. According to the different intensities of change, (very) large negative changes were used as proxy for complete tree cover loss, and medium negative changes were used as potential areas for degradation. The final results were further calibrated based on the socioeconomic results.

The time series Landsat data were created automatically in the SEPAL platform (https://sepal.io/, accessed on 15 July 2022). SEPAL was also used for the processing of the algorithm itself. The computer-intensive process analyzed about 980 Landsat images relating to the AoIs in northern Uganda. The validation of the maps was carried out using field data and the very high spatial resolution imagery Digitalglobe (https://discover.digitalglobe.com/, accessed on 15 July 2022) satellite images provided by UNITAR.

The biomass stock change analysis for the western and southwestern AoIs used different remote sensing techniques to assess changes in both tree cover and biomass. The Global Forest Change data set [22] was used to compute statistics on tree cover loss from 2001 to 2018, within the 5 km and 15 km buffer zones from the refugee settlement boundaries. The data on tree cover loss were overlaid with refugee and host community population data in order to explore potential relationships. In addition, changes in biomass stock between 2000 and 2017 were assessed based on the LULC changes, using the national LULC maps for 2000 and 2017 which were provided by the NFA. Biomass stock for each LULC class for the western and southwestern AoIs (Table 3) was assigned to the national LULC maps. These two maps were then overlaid, and biomass stock changes were calculated by subtracting the pixel values in the 2000 biomass stock map from those in the 2017 map.

The legends of both the 2000 and 2017 maps contain 13 LULC classes. Three of these classes (THF well-stocked, THF low-stocked, and woodland) are considered to be natural forest. The remaining classes are considered to be "other land".

### 3. Results

#### 3.1. Assessing Woodfuel Consumption

The household survey in the sampling hosting areas in northern Uganda (Adjumani and Yumbe Districts) revealed that the average woodfuel consumption per person in a refugee household is lower than that of a host household (Table 5). However, in the two sampled settlements in western and southwestern Uganda, refugees use more woodfuel than host communities. This is because, in these settlements, a greater proportion of refugees than hosts cook with charcoal. The figures provided in Table 5 represent the average woodfuel consumption, expressed as kilogram per person per day (kg pppd).

**Table 5.** Refugee and host woodfuel consumption in the sampled target refugee hosting areas in northern and southwestern Uganda.

| AoI | Population Using Firewood (%) [a] | Firewood Consumption (kg pppd) [b] | Population Using Charcoal (%) [a] | Charcoal Consumption (kg pppd) | Total Woodfuel (kg pppd Wood Equivalent) [c] |
|---|---|---|---|---|---|
| Refugees—Adjumani | 94.3 | 1.73 | 25.3 | 0.25 | 1.95 |
| Refugees—Yumbe | 98.9 | 1.57 | 8.0 | 0.28 | 1.66 |
| Hosts—Adjumani | 98.8 | 2.14 | 7.2 | 0.27 | 2.21 |
| Hosts—Yumbe | 96.5 | 2.13 | 4.7 | 0.25 | 2.11 |
| Refugees—Kyaka II | 31.5 | 0.9 | 77.5 | 0.6 | 2.6 |
| Refugees—Kyangwali | 75.5 | 2.0 | 35.4 | 0.7 | 2.7 |
| Hosts—Kyaka II | 78.8 | 1.6 | 22.0 | 0.9 | 2.2 |
| **Hosts—Kyangwali** | **92.5** | **2.2** | **16.0** | **0.7** | **2.6** |

Note: [a]. Since multiple responses were permitted in the cooking fuel question, the sum of percentages for any location may exceed 100 percent. [b]. Kilograms of firewood per person per day are expressed on an air-dry basis. [c]. Total woodfuel consumption takes into account the rate of consumption of both firewood (expressed on an air-dry basis) and charcoal (expressed in firewood-equivalent, assuming a conversion efficiency of 20 percent).

*3.2. Total Refugee Woodfuel Consumption*

The April 2019 refugee population data in northern Uganda suggest total woodfuel consumption of 421,019 metric tons per year (t/yr) in firewood equivalent (Table 6).

**Table 6.** Estimated total woodfuel consumption by the refugee population in northern Uganda.

| Settlement | Refugee Population (2019) | Total Woodfuel Consumption (t/yr) |
|---|---|---|
| Bidibidi | 225,808 | 149,262 |
| Imvepi | 57,758 | 38,178 |
| Rhino extension—Omugo | 24,533 | 16,217 |
| Agojo | 6661 | 4403 |
| Ayilo I | 23,837 | 15,757 |
| Ayilo II | 13,722 | 9070 |
| Boroli I/II | 14,841 | 9810 |
| Maaji I | 518 | 342 |
| Maaji II | 16,174 | 10,691 |
| Maaji III | 14,947 | 9880 |
| Nyumanzi | 39,505 | 26,113 |
| Pagirinya | 35,803 | 23,666 |
| Palorinya | 119,587 | 79,049 |
| Palabek | 43,238 | 28,581 |
| **Total** | **636,932** | **421,019** |

Note: Woodfuel is expressed on an air-dry basis.

Host population within the 5 km buffer zone could not be estimated due to close distance between these settlements and overlapping areas among the buffers.

Based on the combined population of refugees and host communities within the 5 km buffer zone of the 4 targeted refugee settlements in western and southwestern Uganda (of which refugees and hosts account for 59 and 41 percent, respectively), the total estimated woodfuel consumption is 558,126 t/yr in firewood equivalent (Table 7).

**Table 7.** Estimated woodfuel consumption in the target refugee settlements and 5 km buffer in western and southwestern Uganda.

| Settlement | Refugee Population (2019) | Host Population within 5 km (2019) | Total Woodfuel Consumption (t/yr) |
|---|---|---|---|
| Kiryandongo | 63,365 | 52,950 | 105,310 |
| Kyaka II | 113,023 | 61,004 | 158,652 |
| Kyangwali | 115,488 | 90,308 | 186,527 |
| Rwamwanja | 70,493 | 48,000 | 107,637 |
| **Total** | **362,369** | **252,262** | **558,126** |

Sources: Refugee population data (refugees and asylum seekers): OPM/UNHCR; host population: WorldPop 2019 [23] (based on UBOS data) [24–28]. Note: Woodfuel is expressed on an air-dry basis.

### 3.3. Assessing the Supply of Woodfuel

Potential supply of woodfuel takes into account the AGB stock and the annual AGB growth from woodland and bushland within 5 km of the settlement boundaries for all of the target settlements (Table 8). Figures 4 and 5 show the biomass stock within the AoIs in northern, western, and southwestern Uganda, respectively, in 2018 and 2017.

**Table 8.** Summary of AGB stock and AGB growth for each refugee settlement within 5 km.

| Settlement | District | AGB Stock (t) | AGB Growth (t/yr) |
|---|---|---|---|
| **Refugee settlements in northern Uganda** | | | |
| Bidibidi | Yumbe | 1,093,157 | 29,214 |
| Imvepi | Arua | 282,189 | 7913 |
| Rhino ext.—Omugo | Arua | 111,908 | 2486 |
| Agojo | Adjumani | 92,674 | 379 |
| Ayilo I | Adjumani | 125,180 | 793 |
| Ayilo II | Adjumani | 98,216 | 890 |
| Boroli I/II | Adjumani | 114,172 | 793 |
| Maaji I | Adjumani | 60,270 | 1244 |
| Maaji II | Adjumani | 197,082 | 11,549 |
| Maaji III | Adjumani | 118,157 | 4553 |
| Nyumanzi | Adjumani | 78,584 | 870 |
| Pagirinya | Adjumani | 51,597 | 364 |
| Palorinya | Moyo | 423,178 | 18,170 |
| Palabek [a] | Lamwo | 404,230 | 6767 |
| **Total** | | **3,250,598** | **85,984** |
| **Refugee settlements in western and southwestern Uganda** | | | |
| Kiryandongo | Kiryandongo | 272,229 | 29,384 |
| Kyaka II | Kyegegwa | 432,578 | 39,394 |
| Kyangwali | Kikuube | 1,436,480 | 83,300 |
| Nakivale & Oruchinga | Isingiro | 650,888 | 80,426 |
| Rwamwanja | Kamwenge | 380,139 | 41,961 |
| **Total** | | **3,172,314** | **274,466** |

Note: [a]. Changes in Palabek consider only the most recent years, 2017–2018. AGB growth rates are taken from the NBS as averages for the agro-ecological zone of the AoI. For the AoIs in northern Uganda, growth rates of degraded woodland and bushland estimated by using correction factors of 0.33 and 0.85, respectively, are derived from the ratio of AGB stock of the degraded to that of the conserved classes. AGB growth rates from the NBS were converted from an air-dry to a dry basis, assuming 18 percent moisture content.

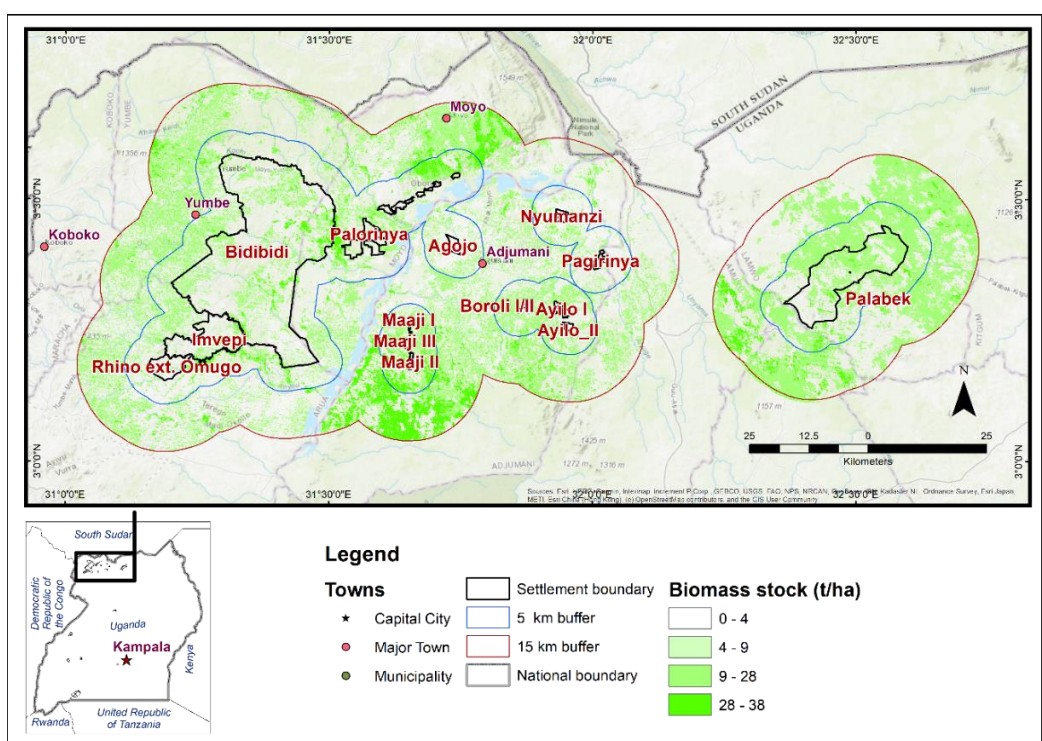

**Figure 4.** Biomass stock within the AoIs in northern Uganda, early 2018. Source: World Bank and FAO, 2020a [29].

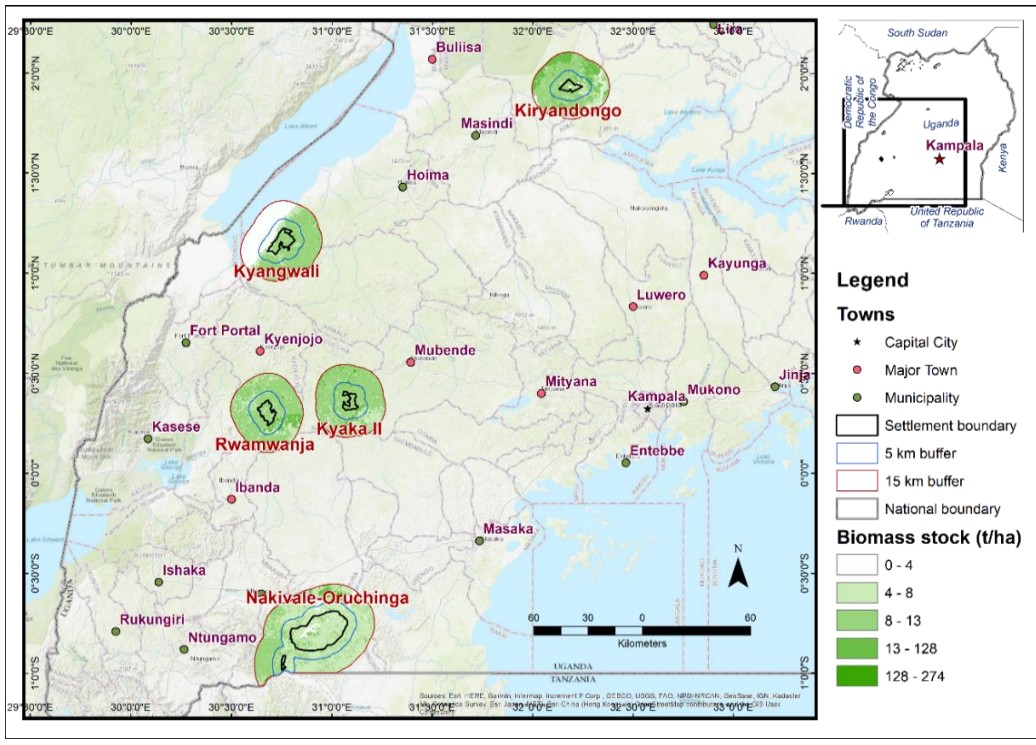

**Figure 5.** Biomass stock within the AoIs in western and southwestern Uganda, 2017. Source: World Bank and FAO, 2020b [30].

### 3.4. Biomass Stock Changes

The map presenting biomass changes in the AoIs in northern Uganda between 2013 and 2018 (Figure 6) shows a reduction in biomass stocks across the whole area, especially northern Bidibidi and around Ayilo and Palabek.

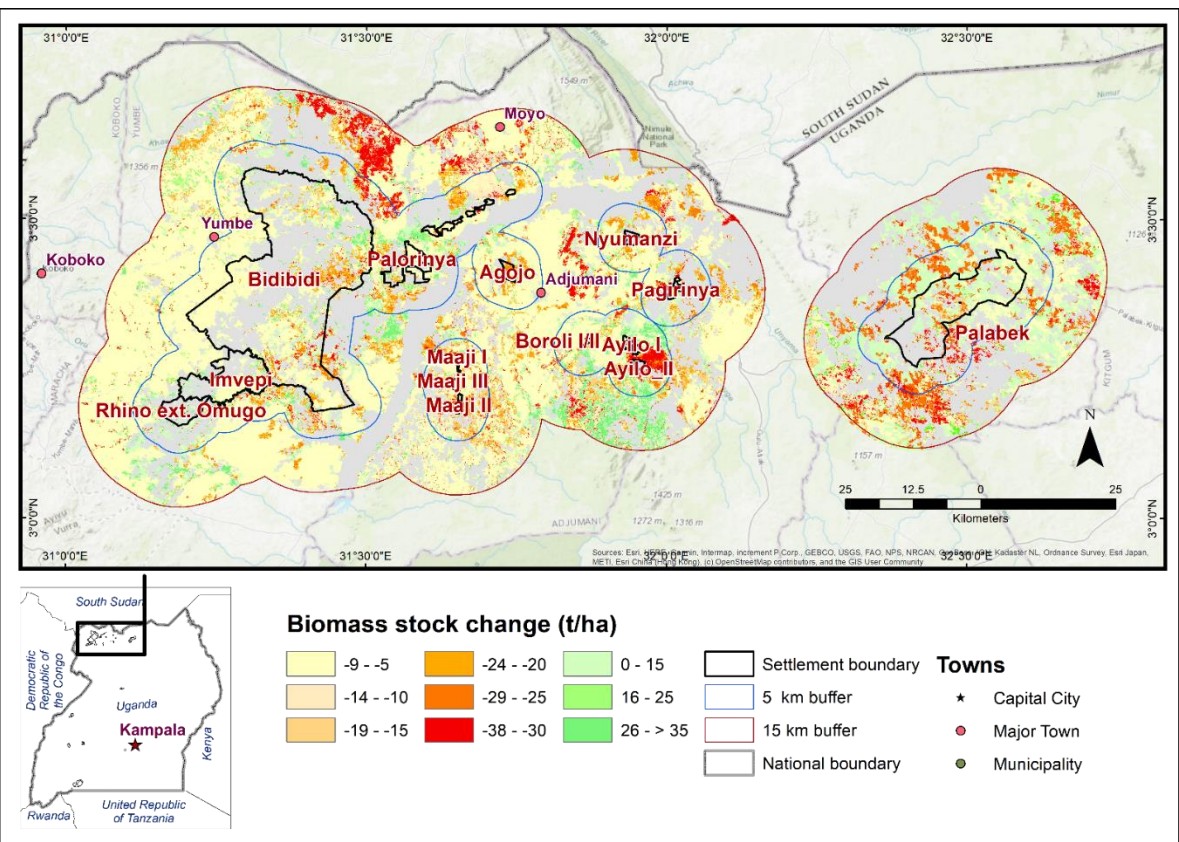

**Figure 6.** Biomass stock changes between 2013 and 2018 within the AoIs in northern Uganda. Source: World Bank and FAO, 2020a [29].

While there is an increase in observed biomass loss, its spatial distribution does not provide strong evidence that this results primarily (or even majorly) from a direct harvesting of woody biomass by refugees. The highest losses are seen in host community areas set back from the settlement boundaries.

According to the results for the 5 km buffer zone in the AoIs in northern Uganda (Table 9), the total tree cover loss between 2010 and 2013 was about 1919 hectares (ha), while degradation covered about 5664 ha (in woodland and bushland, including the areas of the settlements themselves). Meanwhile, from 2014 to 2018, there were 34,112 ha of loss and 29,604 ha of degradation. Total biomass loss accounts for the total loss, including the loss from degraded land. The biomass factor used to compute biomass loss in degraded land is taken as the difference between the biomass factors for conserved woodland (38 t per ha) and degraded woodland (25.3 t per ha), which is 12.7 t per ha. Similarly, for the bushland class, it is the difference between the biomass factors for conserved bushland (27.8 t per ha) and degraded bushland (3.9 t per ha), which is 23.9 t per ha. The overall picture indicates a significant increase in loss and degradation, not only within the 5 km buffer near the refugee settlements, but also in the extended 15 km buffer from their boundaries.

Table 10 highlights the total degradation and loss, including partial loss, in degraded bushland and woodland in the AoIs in northern Uganda. The settlements most affected by major changes in woodland, bushland, and cropland can be noted by comparing the total loss and degradation within the 5 km buffer zone from the boundaries of each settlement (plus the areas of the settlements themselves) over the two periods.

**Table 9.** Loss and degradation (ha) and biomass (AGB) changes in selected land cover classes within 5 and 15 km of the refugee settlement boundaries in northern Uganda.

| Loss and Degradation | 5 km Buffer | | | | 15 km Buffer | | | |
|---|---|---|---|---|---|---|---|---|
| | 2010–2013 | | 2014–2018 | | 2010–2013 | | 2014–2018 | |
| | Total Area (ha) | AGB Stock (t) | Total Area (ha) | AGB Stock (t) | Total Area (ha) | AGB Stock (t) | Total Area (ha) | AGB Stock (t) |
| Loss in woodland | 157 | 5961 | 3288 | 124,950 | 536 | 20,358 | 9253 | 351,614 |
| Loss in bushland | 703 | 19,532 | 6998 | 194,543 | 1428 | 39,696 | 14,015 | 389,624 |
| Loss in cropland | 1060 | 10,521 | 23,826 | 236,591 | 2141 | 21,255 | 54,311 | 539,306 |
| **Total loss** | **1919** | **36,015** | **34,112** | **556,084** | **4104** | **81,309** | **77,579** | **1,280,544** |
| Degraded woodland | 1425 | 36,088 | 10,558 | 267,427 | 4073 | 103,164 | 25,872 | 655,341 |
| Degraded bushland | 4240 | 16,704 | 19,047 | 75,044 | 8797 | 34,660 | 38,787 | 152,822 |
| **Total degradation** | **5664** | | **29,604** | | **12,870** | | **64,660** | |
| Loss in degraded woodland | — | 27,169 | — | 201,336 | — | 77,668 | — | 493,381 |
| Loss in degraded bushland | — | 44,728 | — | 200,942 | — | 92,809 | — | 409,207 |
| Total loss from degraded land | — | **71,897** | — | **402,277** | — | **170,477** | — | **902,588** |
| **Total biomass loss** | **107,912** | | **958,361** | | **251,786** | | **2,183,132** | |

**Table 10.** Summary of degradation and loss (in ha) per settlement within 5 km in northern Uganda.

| Settlement | 2010–2013 | | | 2014–2018 | | |
|---|---|---|---|---|---|---|
| | Degradation (ha) | Loss (ha) | % Loss and Degradation | Degradation (ha) | Loss (ha) | % Loss and Degradation |
| Bidibidi | 1916 | 646 | −1.6 | 12,555 | 9895 | −13.9 |
| Imvepi | 307 | 193 | −1.2 | 3223 | 3682 | −16.5 |
| Rhino ext.—Omugo | 90 | 35 | −0.5 | 1876 | 1237 | −13.6 |
| Agojo | 173 | 92 | −1.8 | 638 | 2921 | −24.4 |
| Ayilo I | 402 | 240 | −4.6 | 1501 | 2073 | −25.6 |
| Ayilo II | 381 | 188 | −4.9 | 1710 | 2252 | −34.0 |
| Boroli I/II | 119 | 134 | −2.3 | 1044 | 532 | −14.2 |
| Maaji I | 116 | 50 | −2.1 | 450 | 738 | −15.1 |
| Maaji II | 228 | 79 | −2.4 | 471 | 435 | −7.2 |
| Maaji III | 638 | 2921 | −30.4 | 860 | 1032 | −16.2 |
| Nyumanzi | 101 | 22 | −0.9 | 1242 | 1254 | −19.3 |
| Pagirinya | 545 | 135 | −4.6 | 615 | 1602 | −15.1 |
| Palorinya | 1728 | 483 | −4.5 | 7771 | 4426 | −24.6 |
| Palabek [a] | 521 | 90 | −0.9 | 1878 | 7727 | −14.1 |
| | Average (%) | | −4.5 | Average (%) | | −18.1 |

Note: [a]. Changes in Palabek consider only the most recent years, 2017–2018.

The maps presenting the biomass stock change from 2000 to 2017 in western and southwestern Uganda (Figure 7) show tree cover and biomass losses at various distances from all the refugee settlements.

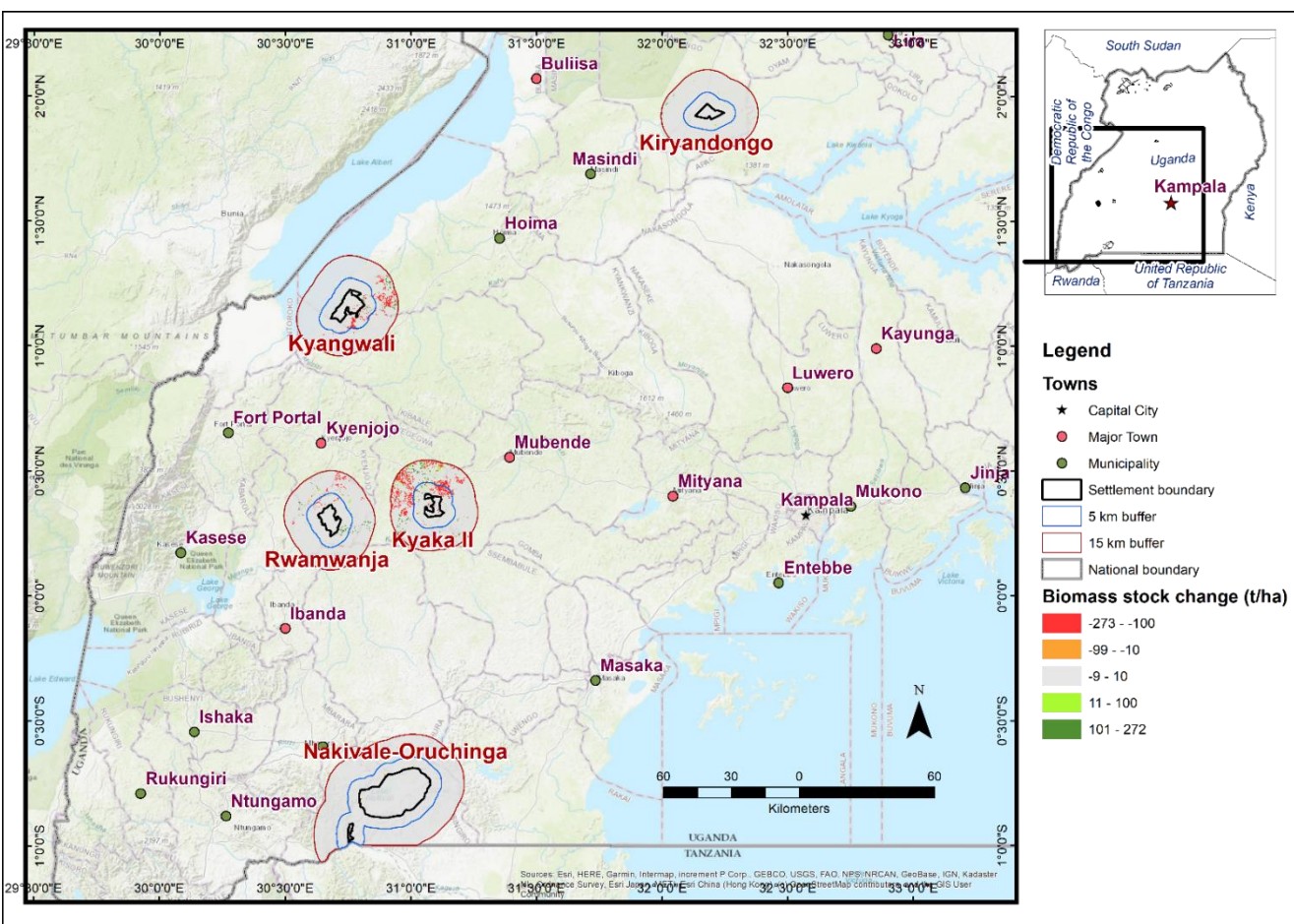

**Figure 7.** Biomass stock changes between 2000 and 2017 within the AoI in western and southwestern Uganda. Source: World Bank and FAO, 2020b [30].

The summary of tree cover loss for the settlement in western and southwestern Uganda in Figure 8 shows that in Kyaka II, Kyangwali, and Nakivale-Oruchinga, tree cover loss was more concentrated in the 5 km buffer than in the 15 km buffer, while at Rwanwanja and Kiryandongo, the opposite was the case. In Kyaka II and Kyangwali, the overall tree cover loss from 2001 to 2018 was close to or greater than 10–13 percent in both the 5 km and 15 km buffers. The lowest tree cover loss, in terms of percentage area, was observed in the Nakivale-Oruchinga AoI, where the presence of trees was already comparatively low.

Among the target AoIs, the highest loss of biomass between 2000 and 2017 occurred within 15 km of the Kyaka II settlement boundary (about 1,673,000 t), followed by Kyangwali (about 1,044,000 t), as shown in Figure 9. Within the 5 km buffer, biomass loss was also highest at Kyaka II (about 358,000 t) and Kyangwali (about 327,000 t). A net gain in biomass was observed only within the 5 km buffer of the Kiryandongo settlement, taking into account the LULC change from 2000 to 2017.

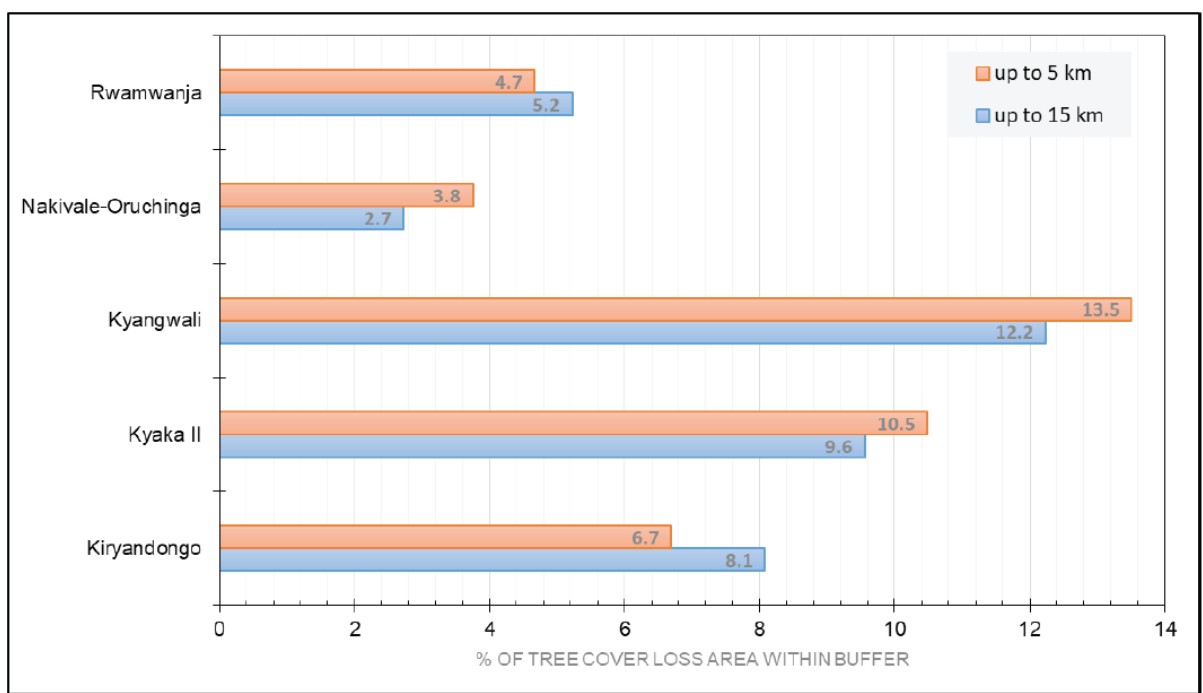

**Figure 8.** Summary of tree cover loss from across buffers in the AoIs (2001–2018) in western and southwestern Uganda. Source: Tree cover loss: Hansen et al., 2013 [22].

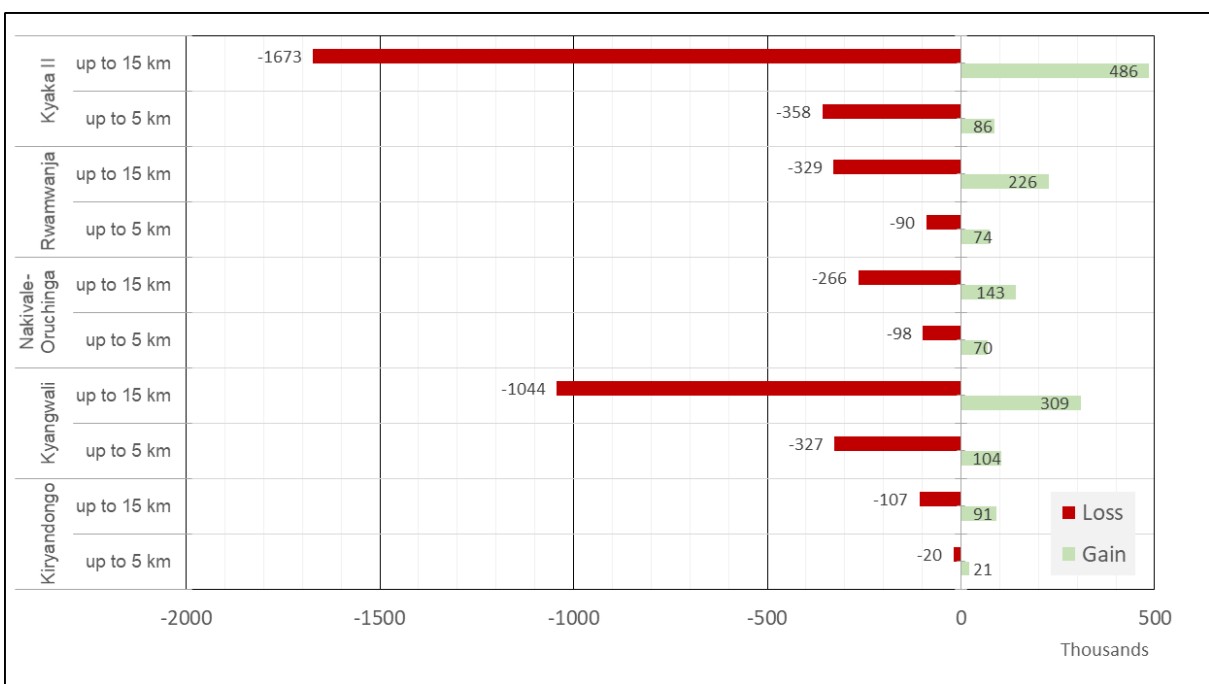

**Figure 9.** Summary of AGB stock changes across the AoIs (2000–2017) in western and southwestern Uganda. Source: World Bank and FAO, 2020b [30]. Note: Loss/gain is associated with LULC changes. For example, if there was a change from higher stock LULC to lower stock LULC, then there was a loss of biomass, and vice versa. Both of these transitions were found in all buffer zones, and respective loss/gain was estimated accordingly.

### 3.5. Linking Woodfuel Demand and Supply

Table 11 shows estimated woodfuel supply and demand for each refugee settlement in northern Uganda, including both firewood and charcoal (the latter converted to a firewood

equivalent). Total woodfuel demand for cooking in the target refugee settlements in northern Uganda is about 34,500 tons of wood per year (on a dry weight basis), based on the refugee population in April 2019. This is about four times the quantity of tree growth within the settlements and the 5 km buffer zone, which could result in an annual biomass deficit of about 8 percent of AGB stock.

**Table 11.** Estimated woodfuel supply and demand within the 5 km buffer zone, northern Uganda.

| AoI | Refugee Woodfuel Demand (t/yr) | AGB Stock (t) | AGB Growth (t/yr) | AGB Change (t/ yr) | Annual Change (%) |
|---|---|---|---|---|---|
| Bidibidi | 122,395 | 1,093,157 | 29,214 | −93,181 | −8.5% |
| Imvepi | 31,306 | 282,189 | 7913 | −23,393 | −8.3% |
| Rhino extension—Omugo | 13,298 | 111,908 | 2486 | −10,812 | −9.7% |
| Agojo | 3610 | 92,674 | 379 | −3232 | −3.5% |
| Ayilo I | 12,921 | 125,180 | 793 | −12,128 | −9.7% |
| Ayilo II | 7437 | 98,216 | 890 | −6548 | −6.7% |
| Boroli I/II | 8044 | 114,172 | 793 | −7251 | −6.4% |
| Maaji I | 280 | 60,270 | 1244 | 964 | 1.6% |
| Maaji II | 8767 | 197,082 | 11,549 | 2782 | 1.4% |
| Maaji III | 8102 | 118,157 | 4553 | −3548 | −3.0% |
| Nyumanzi | 21,413 | 78,584 | 870 | −20,543 | −26.1% |
| Pagirinya | 19,406 | 51,597 | 364 | −19,042 | −36.9% |
| Palorinya | 64,820 | 423,178 | 18,170 | −46,651 | −11.0% |
| Palabek | 23,436 | 404,230 | 6767 | −16,669 | −4.1% |
| **Total** | **345,236** | **3,250,598** | **85,984** | **−259,251** | **−8.0%** |

Note: Woodfuel demand was converted from an air-dry to a dry basis, assuming 18 percent moisture content. Estimate of annual AGB loss takes into account household woodfuel demand based on April 2019 refugee population, though field observations highlighted other demand for woody biomass for construction, as well as energy for commercial and economic activities, agricultural activities, and losses to fire.

With reference to the target area in western and southwestern Uganda, based on the combined population of refugees and host communities within the 5 km buffer zone of the four western refugee settlements, taking into account only the woody biomass from the 5 km buffer zones, and assuming that woodfuel demand is met only with this biomass, there is an annual deficit equivalent to 11 percent of AGB stock (Table 12).

**Table 12.** Estimated woodfuel demand and supply within the 5 km buffer zone, western and southwestern Uganda.

| AoI | Refugee and Hosts Woodfuel Demand (t/yr) | AGB Stock (t) | AGB Growth (t/yr) | AGB Change (t/yr) | Annual Change (%) |
|---|---|---|---|---|---|
| Kiryandongo | 92,902 | 272,229 | 29,384 | −63,518 | −23 |
| Kyaka II | 128,189 | 432,578 | 39,394 | −88,795 | −21 |
| Kyangwali | 164,446 | 1,436,480 | 83,300 | −81,146 | −6 |
| Rwamwanja | 86,566 | 380,139 | 41,961 | −44,605 | −12 |
| **Total** | **472,103** | **2,521,426** | **194,039** | **−278,063** | **−11** |

Sources: Refugee population data (refugees and asylum seekers): OPM/UNHCR; host population: WorldPop 2019 [23] (based on UBOS data) [24–28]. Note: AGB is estimated on a dry basis and includes the biomass of the settlements themselves. Woodfuel demand was converted from an air-dry to a dry basis, assuming 18 percent moisture content of firewood, using the original data (Table 5). AGB growth rates from the National Biomass Study (NBS) (Forest Department 2002) are national averages, converted from an air-dry to a dry basis, assuming the same 18 percent moisture content.

## 4. Discussion

The refugee influx has led to an increase in the rate of degradation and tree loss with accelerated land cover changes. Deforestation and forest degradation are not new phenomena in Uganda, and the refugee presence has added to existing pressures on the environment due to increased demand for wood as cooking fuel. Competition for available resources could become a source of tension between the refugees and host communities.

Both refugee and host community households depend on woodfuel for cooking. The three-stone open fire is still widely used for cooking, and both refugees and host communities have a tradition of building improved mud-stoves from locally available materials. Basic improved stoves made of metal and clay are also seen. Within the refugee settlements, greater diversity in cooking devices exists than in host communities, and a higher proportion of refugee households use improved stoves. Modern prefabricated stoves are also available, but they are too expensive for most refugees and locals.

In the refugee hosting areas in northern Uganda, average woodfuel consumption per person for cooking is about 20 percent higher in host communities than in the refugee settlements. In western settlements, refugee households use less energy than the hosts when using the same fuel type, but they consume more overall in terms of "wood-equivalent", as they are more likely to cook with charcoal.

Total wood demand for cooking in northern refugee settlements is about 345,000 metric tons (t) per year, and 475,000 t (on a dry basis) in the four settlements of Kyangwali, Kyaka II, Kiryandongo, and Rwamwanja in western and southwestern Uganda. Construction, commercial and economic activities, agricultural activities, and losses to fire further contribute to the total demand for woody biomass.

Comparing woodfuel demand with potential biomass supply within a 5 km buffer suggests an annual deficit equivalent to 8 percent of AGB stock in the northern settlements, and 11 percent of AGB stock in the western and southwestern settlements. Yet, actual recorded tree cover loss and changes in land use and land cover do not always reflect these expected losses. In other words, the expected loss of AGB stock based on the woodfuel consumption data is not, in fact, fully observed to the extent of the observed ABG biomass stock from the remote sensing analysis.

Based on the results within the 5 km buffer of the AoIs in northern Uganda, biomass loss increased from 107,912 t to 958,361 t in the periods between 2010 and 2013 and 2014 and 2018, respectively. Additional biomass loss increased in the extended 15 km buffer over the two periods of analysis—though the latter more likely reflects ongoing degradation by host communities rather than a direct harvesting of wood by refugees.

These discrepancies could be due to a partial supply of fuel (especially charcoal) from further away, and potentially by absenteeism among both refugees and hosts. In addition, inaccuracies of the national LULC maps, the biomass factors applied for each LULC class, and the degradation/loss maps obtained from BFAST and the Global Forest Change dataset could have contributed to some discrepancies observed between the data collected on the ground and the data from the remote sensing analysis. Additionally, the estimated daily woodfuel consumption per person in the households sampled, which was then extrapolated for the total woodfuel demand of all the refugee settlements in the AoIs, might present some deviations.

On the other hand, field observations highlighted numerous other demands for forest products for construction, energy for commercial and economic activities, agricultural activities, and losses to fire, which further contribute to the overall demand for woody biomass and are not included in the calculations in this study.

The observed changes in tree cover and AGB require a coordinated response that involves both host communities and refugees in an effort to achieve more sustainable management of forest resources across the hosting refugee areas.

The assessments recommend the following interventions to improve forest resource management, ensure access to woodfuel, and contribute to building livelihood resilience for both refugee and host communities:

1. Development of agroforestry systems on household plots and farmland, where trees and woody perennials are interplanted along boundaries and/or with crops for energy, food, and fodder. This intervention should target the residential plots assigned to refugees and the cultivated fields of both host and refugee communities surrounding refugee settlements.
2. Establishment of woodlots for energy and other purposes, such as building material. This intervention should target areas owned by host communities and individuals, degraded areas managed by the NFA, and areas assigned for refugee settlements.
3. Rehabilitation of degraded forests using both natural and assisted regeneration. This intervention should target areas owned by host communities and individuals, degraded areas managed by the NFA, and areas assigned for refugee settlements.
4. Enhancement of energy efficiency to reduce the demand for woodfuel through more efficient cooking practices and charcoal production techniques. This intervention should target both host and refugee populations.

These recommended interventions should be prioritized in the settlements and the surrounding areas, which have major negative impacts on the environment as a result of poor access to modern energy for cooking. These interventions should be coordinated as part of an integrated energy and environment program with sufficient institutional capacity, resources, and technical expertise to undertake more in-depth analysis, carry out monitoring and evaluation, support systematic efforts to promote the interventions across the associated host communities, and ensure sound learning, sharing, and interaction with other programs of a similar nature in Uganda. This will ensure that these measures do not take place in isolation, nor in a scattered or short-term manner.

## 5. Conclusions

The refugee population in Uganda has increased dramatically following the settlement of over 1.2 million refugees since 2014, and this presents a risk of competition with host communities for the use of natural resources such as land, water, and wood, which can ultimately cause deforestation and/or environmental degradation. Impacts on the surrounding environment of refugee settlements resulting from the collection and production of firewood and charcoal can be lasting and damaging.

The livelihoods of refugee and host households are highly dependent on forests and other woodlands as primary sources of woodfuel for cooking. This assessment indicates a steady increase in degradation and vegetation loss over the hosting area, and map comparisons reveal increased land cover changes in the woodland and bushland. The areas within the settlements and the buffer zone of 5 km around their boundaries have been subjected to changes after the arrival of refugees, and, in some of the target settlements, competition for the available resources could become a source of tension between the refugee communities and hosts living in their immediate surroundings.

Planning for the sustainable supply of energy plays a crucial role in minimizing environmental impacts and conflicts with host communities over the use of natural resources. Dedicated woodlots can provide a sustainable supply of woodfuel, and restoration interventions on degraded land enhance availability and productivity of forest products (wood and non-wood forest products) as well as ecosystem services. Agroforestry interventions, along with a more efficient use of energy for cooking and charcoal production, can reduce these environmental impacts.

It is expected that refugee and host communities will continue using firewood and charcoal for the foreseeable future as their primary sources of energy. Therefore, responsible planning for sustainable harvesting, production and use of firewood is crucial for ensuring energy access and, in turn, building resilience in the refugee-affected areas and contributing to food and nutrition security.

**Author Contributions:** A.G., N.G.B., R.J., L.H., S.W., T.L.-J. and Z.X. contributed to the drafting of the article, its review, and the writing of the final version. All authors have read and agreed to the published version of the manuscript.

**Funding:** This research received no external funding. The article processing charge (APC) was funded by FAO.

**Institutional Review Board Statement:** Not applicable.

**Informed Consent Statement:** Not applicable.

**Data Availability Statement:** Not applicable.

**Acknowledgments:** The authors acknowledge the support of the World Bank for the development of the original assessments conducted in 2018 and 2019 on the refugee hosting areas in Uganda.

**Conflicts of Interest:** The authors declare no conflict of interest.

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
