# Peer review of "Woodfuel Consumption in Refugee Hosting Areas and Its Impact on the Surrounding Forests—The Case of Uganda"

_forests, doi:10.3390/f13101676_

Round 1
Reviewer 1 Report
The article addresses the impact of woodfuel consumption in refugee hosting areas on surrounding forests in Uganda. The manuscript topic is interesting and very relevant for the study sites. The study integrates woodfuel consumption data, AGB field inventory information, and degradation/loss area identification using remote sensing images or LULC maps to estimate the relationship among these three sources of information. The authors conclude that the pressure generated by the need of firewood and charcoal by both host and refugee communities will continue to cause the loss and degradation of the remaining forested areas in the studied areas.
General comments
The article's topic is interesting; however, its soundness is hindered by three main points that need to be corrected in a future version.
1. The reference list contains only 13 works and most of them are technical reports. I strongly suggest including more references, especially related journal articles, in both the Introduction and Discussion sections. This will not only make the article more robust, but also give a broader context to discuss the results in the light of other previously published studies.
2. The manuscript is hard to follow since several key methodological aspects are not mentioned or justified, and practically all the results are not shown homogeneously for Northern, Western and Southwestern Uganda. Please go to the particular comments section for a detailed description of key missing methodological aspects. It is unclear if the results for each region are comparable among them or not, since they followed slightly different methods and are always shown separately. I consider that results for all the AOI could be shown together and then, the possible implications of the different followed methods could be discussed in the Discussion. Finally, the authors should explain why they used different methods to estimate AGB in northern, southwestern and western Uganda.
3. The Discussion section is missing several key points. In the following lines I suggest several topics in which the article could deepen its discussion: 1) forest degradation/loss patterns that are similar for the three regions, as well as possible differences, 2) potential limitations of the study or certain parts of the method (e.g., of using BFAST to estimate forest degradation/loss, evaluating degradation/loss using national LULC maps or the method used to estimate wood demand/supply), 3) possible differences in the estimations done for the different AOI due to the use of different methods, 4) compare the reported annual net loss rates with other regions in Uganda or even a national estimate.
Particular comments
Lines 25-26, 28-30, 30-33, 40-44, 44-47. These lines need at least a citation.
The Introduction is missing a paragraph about previous studies evaluating the demand/supply of woodfuel. Please consider adding this paragraph with its corresponding references. Additionally, it would be desirable to mention certain aspects of mapping forest degradation/loss using remote sensing images and time-series algorithms like BFAST, as well as LULC maps. Finally, the Introduction should introduce certain key terms for the paper such as forest loss or deforestation, and forest degradation.
Line 73, 73. The Figures show a 15 km buffer that is not mentioned in the study, please remove this feature from the map and legend.
Lines 82-89. Please mention separately the sample size for refugee and host-community households, as well, as how many observations were collected in the north, west and southwest AOI.
Lines 91-92. Can the authors show a map of the LULC classes in the study region. It might give a better characterization of the study area. Please also consider adding a description of the difference between bushland and woodland.
Lines 92-93. What is the definition of woodland and bushland depleted?
Lines 101. Are degraded woodland and degraded bushland additional LULC classes? Please further explain if these consist of other LULC classes or how were they identified.
Line 103. Depleted is the same as degraded? If that is the case please consider only using the term degraded.
Line 103. How was degradation identified. Please consider giving further details.
Line 104-105. There are several aspects of the plots that are not clear. For example,
a. How many plots were sampled for each LULC class?
b. Was there a selection of 95 plots and then only 67 were sampled? Why was that and how was this number determined? For example, line 100 mentions that 15 plots per class were assigned; however, 5 classes * 15 plots = 75.
c. How were these plots selected?
d. What was the target sample number for each stratum?
Line 109. I believe the correct reference for Chave et al. [6] is 2015, not 2014. Please consider giving more details about the variables sampled in the field plots. Chave et al. allometric equations consider DAP (diameter at breast height), species identity and height. Were these variables sampled in the field? Also mention that the data was extrapolated to 1 ha. Finally, consider describing most of the text in Figure 3 as part of the manuscript text, instead of the figure.
Line 110-111. Please give more details about what these R scripts do. Any reader should understand how the stocks were calculated.
Line 112. Shrub biomass correspond to another LULC or certain trees in the field sampling? Please clarify this, also mention how were shrubs (as trees or LULC) defined or identified.
Line 114. Table 2. Please consider changing "intact" for "conserved" or a similar term that do not imply that it has not been "touched", since proving that is often impossible.
Lines 116. Is this the same National Biomass Study (NBS) or another one? Please confirm. If it is the same, please use the same name, if not, please mention why the NBS data was not used.
Line 118. What does harmonization means in this context? Why are other LULC presented here instead of the five presented for Northern Uganda (woodland, bushland, cropland, woodland depleted and bushland depleted).
Line 120. What was the purpose of calculating the biomass growth? Please mention it here
Lines 125-126. What does THFL and THF stand for? Please add its meaning either in the table or in the table captions.
Lines 128-133. This paragraph is missing several key details. For example, why were the before and after Sudan crisis periods selected as 2010-2013 and 2014-2018, if the data used to quantify biomass stock change corresponded to 2010 and 2015. Please further explain this. Also, how was this information combined with the BFAST results. In addition, no details are provided about how the BFAST analysis was performed. Please consider including the following information: type of imagery used (e.g., Landsat), processing level (e.g., surface reflectance), spectral or vegetation index used to run BFAST (e.g., NDVI), span of the historic and monitoring periods, change magnitude threshold to identify the change class, the description of how degradation and loss were differentiated, if the BFAST results were verified and what was its accuracy, and if any additional post processing was applied.
Lines 134-135. Please add details about how the biomass stock maps were calculated using the LULC maps. I assume that this was done using Table 3 values, but please mention it. Additionally, please mention the spatial scale of these LULC maps. The same goes for the LULC maps mentioned in Line 131.
Line 142-148. Please consider testing for a significant difference using a statistical test. This will give more robustness to the claims mentioned in this paragraph.
Line 151. Even though multiple responses were permitted in the interviews, why was the data not scaled to sum 100 %. I suggest doing so, since it will help interpret the results.
Line 163. Please explain with more detail the agro-ecological zones. Is there a reference for this data. Please consider mentioning it in the Methods section.
Line 164-165. Please further explain these different social and ecological contexts.
Lines 157-167. This sounds more like Methods. Consider moving it to the Methods section.
Line 167-169. This population's woodfuel consumption includes refugees and host-communities inhabitants? Please clarify.
Lines 205-207. Did the biomass stock change estimates (using remote sensing or LULC maps) agreed with the calculated woodfuel demand? This might be an interesting aspect to further discuss the results of the study or an indirect way to estimate the magnitude of other unmeasured wood uses (e.g., construction, commercial, agricultural, economic activities or even fires; lines 270-271).
Line 208. Why the maps for Western and Southwestern Uganda are not shown? Please consider adding them.
Line 214. Why the AGB stock change for Northern Uganda is not shown. Consider adding them to this figure or showing them in a separate figure.
Line 227. What is the meaning of HH?
Line 228-230. This should be placed in the Discussion section.
Lines 237-238. It is not clear how this 11 % annual deficit was calculated. The only way I can obtain 11 % is by dividing 281 091 (AGB loss) / 2 521 426 (AGB stock) = 11.15 %. However, this calculation does not account for the AGB growth (194 039). Is this a mistake in the calculation? If not, please justify this decision.
Line 251. Deforestation and forest degradation concepts appear until the discussion. Please consider adding some lines about these concepts in the introduction section.
Lines 250-292. These paragraphs need references.
Please check the cross references to certain tables and figures. Some of them show the "Error! Reference source not found".
Please consider homogeneizing the maps style. For example, Figure 1, line 73 and Figure 1, line 78. Please also check the figures and tables numbering.
Lines 273 - 286. Please consider further discussing the limitations of the study.
Reviewer 2 Report
1. Double-check the references in the manuscript.
2. Add more references, 13 references don't support the research.
3. The conclusions aren’t supported by the results. The conclusion and discussion section are mixed. Please review both sections.
4. Double-check the reference section.
Author Response
- Double-check the references in the manuscript.
Response 1: Done
- Add more references, 13 references don't support the research.
Response 2: Done
- The conclusions aren’t supported by the results. The conclusion and discussion section are mixed. Please review both sections.
Response 3: Sections reviewed.
- Double-check the reference section.
Response 4: Done.
Reviewer 3 Report
The subject sounds interesting and contemporary, but the following aspects need to be improved:
Introduction
- The introduction is too general and less focused. It does not have a solid scientific foundation - there are too few bibliographic sources studied and mentioned by the authors; I recommend the identification of some scientific articles in which the topic was addressed and the insertion of the corresponding information, respectively citing the sources, to improve the introduction
- The purpose and objectives of the research or the research hypotheses are not well highlighted
Methodology
- I recommend a more detailed description of the research tools, respectively of the data collected by questionnaire and interview and their analysis
- Resolving errors regarding cited sources
Results
- A brief interpretation of the data from each table/figure inserted in the text is required (below each table/figure)
Discussions
- I did not see any discussion about the implications of the study or about the theoretical contribution of the research, nor were other works from the specialized literature mentioned in correlation with the results obtained
Conclusions
- No information is mentioned about the novelty of the study, the need to carry out the study or whether similar studies have been carried out before
I hope the authors will positively accept these constructive suggestions as a way to move this research forward.
Round 2
Reviewer 1 Report
This new version corrected most of the issues present in the previous version; however, there are a few minor aspects that I suggest correcting before its publication.
Line 159. There are still issues with the total number of plots and its distribution. Lines 148 - 149 state that biophysical data was collected for 5 LULC classes. Line 161 mentions that 15 plots per class were assigned. Finally, line 159 mentions that a total of 95 plots were identified and targeted. There is a numeric disagreement among these values. 5 LULC classes * 15 plots = 75 plots. Please make sure these values agree among them. More LULC were included and then definition of the 95 plots? Or more plots were asssigned by LULC class?
The cross references are still nor properly referenced. See for example line 272.
Please homogeneize results presentation in Tables 6 and 7. Table 6 shows only population and woodfuel consumption of refugees, while Table 7, includes host population and woodfuel consumption is aggregated by refugee and host population. Please choose a similar format for both tables.
Line 338. Point worth discussing: These losses might not be related to the refugees but rather host communities? I strongly suggest mentioning this point in the discussion.
Table 9 is not entirely clear. In some rows, the values shown seem to correspond to total area and AGB stock losses (Loss in woodland / bushland / cropland), while in others it seems to correspond to total area and AGB stock (degraded woodland / bushland; thus the AGB stock loss is shown in separate rows). Please reorganize this Table and standardize the meaning of each column. Finally, if total area and AGB stock losses are being shown please add the term "loss" in the headers.
Lines 370 - 376. Please mention that these results are for western and southewestern Uganda.
Table 11. Change "Woodfuel demand (refugees)" for "Refugee woodfuel demand". Instead of loss/gain I suggest using change and clarify that negative values represent loss and positive ones, gain in the table's caption.
Line 455-458. The degradation/loss maps obtained using BFAST and the Global Forest Change dataset also contain errors. Thus, this is another source of error that might explain the inconsistencies. Please at least mention it briefly. There are several previous studies that report this aspect. Please add related references to this part.
Lines 465-467. I am certain that there are several previous studies that could be mentioned and cited in this paragraph about inconsistencies of field and remote sensed evaluations, as well as the absenteeism. This will give the reader a broader vision that similar findings have been reported in other studies.
Lines 475. So the actual estimations of wood demand could be higher than the ones estimated here? This might be a point worth highlighting.
Lines 479. By taking a look to Table 11 and 12, one can see that the annual change rate is very different by AoI. This aspect is not mentioned in the Discussion and I think it would be very relevant, since particular AoIs should be priorized to implement the suggested interventions.
Finally, I suggest using the same format for all the tables and figures.
Reviewer 3 Report
The authors made substantial changes to the work, according to the recommendations received. Consequently, I consider that the article can be published in Forests.
Author Response
We are very grateful to the anonymous reviewer for the comments that helped improve the paper.